# Exploring the Influence of Smart Product Service Systems on Enterprise Competitive Advantage from the Perspective of Value Creation

Linna Hou , Juning Su * and Yu Ye

School of Economics and Management, Xi'an University of Technology, Xi'an 710054, China; houlinnaie@163.com (L.H.)
* Correspondence: sujn@163.com

**Abstract:** With the continuous development of information and communication technology, the development of smart product service systems (smart PSS) has become a crucial approach for enterprises to establish a competitive advantage. However, there is still a lack of clarity regarding the impact process of smart PSS on competitive advantage. This paper aims to explore the impact mechanism of smart PSS on competitive advantage from the perspective of value creation, using an intelligent connected vehicle supplier as a case study. The findings reveal that the composition of smart PSS, including smart technology, smart products, and smart services, influences competitive advantage through the value creation process, which involves relationship construction, relationship operation, and value release. Under the smart PSS environment, changes occur in subject relationships, resources, and key elements. In the context of smart PSS, deep cooperation between enterprises and partners focuses on ecological advantages, while general cooperation emphasizes benefit advantages. This research provides valuable insights into the impact of smart PSS composition on competitive advantage and offers reference values for organizations to establish competitive advantage objectives.

**Keywords:** smart PSS; value creation process; competitive advantage

## 1. Introduction

With the rapid development and application of intelligent technologies such as the Internet of Things, big data, artificial intelligence, and 5G communication, the integration of products and services has become more digital, intelligent, and interconnected [1]. This has resulted in an increasing number of enterprises transitioning from PSS to smart PSS. smart PSS has the characteristics of self-perception, self-monitoring, self-control, and self-decision-making [2]. It can collect real-time data on products, users, and the surrounding environment and conduct cloud processing and analysis to identify potential customer needs. Enterprises aim to meet customers' personalized and time-varying needs, as well as real-time and active service goals. Additionally, smart PSS sets new standards for organizations to achieve competitive advantage and changes the production activities, cooperation modes, and resource allocation in the value chain [3]. Therefore, it is crucial to reveal the impact of smart PSS on competitive advantage.

To build their competitive advantage, enterprises are considering how to use smart PSS. Some enterprises have deconstructed their smart PSS and clearly defined the value-added process, including intelligent products, technologies, and services. For example, SANY Heavy Industry Company relies on the "root cloud platform" to install data detectors on construction machinery, develop service apps, and provide various industrial machinery indexes and electronic services, which has brought high market benefits and ecological advantages. Xiaomi has become one of the top three smart home system manufacturers in China by integrating relevant businesses and building smart ecological home systems based on smartphone user groups. However, some enterprises, such as LeTV, that invested

heavily in the research and development of intelligent connected vehicles eventually faced capital chain interruption and enterprise bankruptcy. These examples show the urgent need to understand how the development of smart PSS affects competitive advantage.

Prior research found that intelligent products, intelligent services, and intelligent technologies promote enterprises to enhance their competitive advantage separately [3–6]. However, the research separated the connection of smart PSS components, which prevented the full utilization of the system's function. As a result, it was difficult to comprehensively analyze the value creation processes, including relationship construction, relationship operation, and value release, and realize the relationship mapping from smart PSS to competitive advantage. To address these issues, this paper discusses the impact of smart PSS on competitive advantage from the perspective of value creation.

The objective of this study is to examine the significance of smart PSS in relation to competitive advantage. By exploring the impact of smart PSS on value creation processes within organizations, this research aims to provide insights into the comprehensive analysis of value creation within the smart PSS framework and its implications for competitive advantage. The findings of this study will contribute to a better understanding of how the development and implementation of smart PSS can affect competitive advantage, enabling businesses to make informed decisions and strategies for leveraging this technology for sustainable success. Ultimately, this research aims to bridge the gap in the existing literature by examining the interconnectedness of smart PSS components and their role in achieving a competitive advantage within the value chain.

## 2. Literature Review

### 2.1. Smart Product Service System

The concept of smart PSS was initially proposed by Valencia [7] and defined as the integration of intelligent products and electronic services through information and communication technology (referred to as ICT technology), which is aimed at satisfying user needs. This definition has been widely accepted by scholars, although some have proposed different interpretations. For instance, Hagen et al. [8] defined smart PSS as the digitalization of the entire product service system, where intelligent products and services should be seamlessly integrated, enabled, and supported by ICT technology. Meanwhile, Zheng et al. [9] believed that smart PSS is an IT-driven business strategy consisting of stakeholders as participants, intelligent systems as infrastructure, intelligent products as media and tools, and electronic services as the key value of delivery. It is committed to meeting individual customers' needs continuously. Research on stakeholders by Wang, J. et al. [10] indicates that external partnerships play an important role in the relationship between internal capabilities and service diversity within a company. Jalil, M. [11] evaluates the impact of government policies on the development of small and medium-sized enterprises. Iinuma, M. et al. [12] describe the creation and dissemination process of new knowledge among companies and internally. Warnakulasooriya, W.H.S. et al. [13] evaluate the factors influencing the product demand of the new generation of customers. Zhang, Z. et al. [14] analyze the requirements of product service systems based on customer demand diversity. Su, J. et al. [15] design service optimization solutions to meet the needs of customers for remanufacturing. Hosaka, T. et al. [16] design online services based on customer network browsing behavior. Chowdhury et al. [17] found that smart PSS has implications for value systems and business models, as well as boundary spanning and dynamic capabilities. While there are varying definitions of smart PSS among scholars, the majority agree that information and communication technology play a significant role in its implementation.

The composition of smart PSS is a crucial topic in academic research, with current studies primarily falling into three categories. Firstly, some focus on describing the composition of smart PSS through the perspective of intelligent technology or products. For instance, Valencia [7,18] believes that smart PSS is supported by intelligent technology, while Zheng et al. [9] argue that smart PSS uses intelligent products as tools and media. Secondly, others describe the composition from the perspective of the relationship between

intelligent technology, products, and services. Hagen et al. [8] propose that smart PSS is the integration of intelligent products and services that are enabled and supported by intelligent information and communication technology. Meanwhile, Chen et al. [5] suggest that intelligent technology and connected products form part of intelligent service systems, which encompass smart PSS. Thirdly, some describe the composition of smart PSS from the perspective of product and service stakeholders, alongside intelligent products and services. Wu [1] proposes that smart PSS comprises three components: stakeholders, intelligent product systems, and intelligent service systems.

Some concepts of smart PSS are described from the perspective of their relationship with PSS. Zhou et al. [19] proposed that the rapid deployment of digital technologies drives the transformation of PSS into SPSS. smart PSS possesses intelligent capabilities and holds enormous potential to create a superior user experience. Yang et al. [20] developed a four-step PSS design method oriented towards service demand, enabling service providers to gradually design specific service orders according to this method. This showcases the design process of industrial SPSS. Moro et al. [21] presented a historical development chart of literature reviews related to PSS, which is divided into four research periods. They also suggested that future studies may focus on digital trends and smart PSS in the coming years. Although scholars have varying opinions on the components of smart PSS, they generally agree that intelligent technology, products, and services are integral parts.

### 2.2. Smart PSS and Competitive Advantage

Scholars have gradually recognized the impact of smart PSS composition on enterprise competitive advantage since its proposal, leading to relevant research. Porter et al. [3,4] discussed how intelligent products affect the organizational structure, data governance, and service methods of enterprises. They studied the establishment of sustainable competitive advantage through smart connected products (SCPs), which differ from smart PSS as it only considers the impact of intelligent connected products on enterprises. In contrast, smart PSS combines services and products to meet consumer needs jointly [22], which has a more significant impact on enterprise competitive advantage. Chen et al. [5] believe that smart PSS's essence is an intelligent service system and collectively refers to services based on intelligent technology and connected products as intelligent services. They also investigated the impact of intelligent services on enterprise innovation and sustainable competitive advantage in the digital era. Furthermore, based on grounded theory, Su et al. [6] examined how intelligent technology promotes the intelligent transformation and upgrading of manufacturing enterprises. Undoubtedly, the impact of smart PSS composition on enterprise competitive advantage is becoming a hot topic among scholars. Langley [23] discovered that PSS and service-oriented business models have tremendous potential for achieving sustainable manufacturing while maintaining competitiveness and fostering innovation. However, to tap into this potential, a more profound understanding of the subtle differences within these business models is required. Dalenogare et al. [24] emphasized that when companies adopt a symbiotic business approach, practitioners must strengthen mutual trust due to the reciprocal dependency of value creation and acquisition. Additionally, their partners can also become competitors in the market. We demonstrate that establishing power structures and developing reciprocal awareness are crucial for successful cooperation in such enterprises. Moreover, we provide similar insights for three other configurations, illustrating the relevance of each configuration in managing ecosystems. As a result, managers can understand how to collaborate with other companies through smart PSS to maximize benefits and influence their relationships. Taking into account the internal relationships and characteristics between products and services, as well as the matching preferences of manufacturers and customers, Luo et al. [25] have developed a configuration method that seeks one-to-one matching between components and services to enhance competitive advantage.

### 2.3. Value Creation

Value creation theory is a vital theoretical foundation for enterprises to build competitive advantages and maximize value [26]. Research on value creation mainly concentrates on four aspects: (1) the source of value creation; (2) the participants involved in value creation; (3) the process of value creation; and (4) the goal of value creation [27]. Several scholars have conducted research on the process of value creation from different perspectives.

Gulati, Dyer, and others [28,29] emphasized the importance of enterprise cooperation in establishing competitive advantage and divided the process of enterprise value creation into three stages: enterprise relationship building, relationship operation, and value release. However, they did not consider how data have replaced products as the core resource in the value creation process with the rapid development of intelligent technology. Consequently, this has brought profound changes to the relationship operation mechanism and value release process. Ding et al. [30] studied the impact of corporate social responsibility on the innovation value creation process and divided the process into three stages: value capture, value conversion, and value realization. Additionally, Wang et al. [31] analyzed the value creation process mechanism of the enterprise product service system from the perspective of social responsibility integration, dividing the process into five core links: value cognition, value proposition, value integration, value communication, and value realization. However, under the fast development environment of intelligent technology, the product service system evolved into smart PSS, and the digitalization of the enterprise value creation process became evident. The impact of intelligent technology on each stage of the value creation process has not been thoroughly analyzed in the aforementioned research. Vial [32] and others discussed the impact of digital technology on the enterprise value creation process, subdividing the process into value proposition, value network, digital channel, realizing agility, and dual flexibility. However, they limited the results of the enterprise value creation process to enhancing agility and flexibility within the enterprise, disregarding the impact of digital technology on enterprise cooperation. In smart PSS, the influence of enterprise partnerships on the process of enterprise value creation and the establishment of competitive advantage is increasingly significant. Enterprises gather data through intelligent products, collaborate with clients and partners, release data value, provide relevant business, and establish their sustainable competitive advantage. Therefore, the value creation process of smart PSS is no longer limited to the internal influence of the enterprise. Instead, it aims to achieve ecological chain coordination through partnership and build an enterprise-led ecosystem. Machchhar et al. [33] conducted a review of the most recent technologies for data collection and utilization in value creation, with a specific emphasis on the convergence of data, PSS, and value. Their research sheds light on the intersection of these domains and provides valuable insights. Rapaccini and Adrodegari [34] integrated the mechanism of value co-creation with service delivery choices. For instance, they emphasized that data-driven self-service should be designed to be easily accessible and user-friendly, while interactive services should strive for frictionless interaction. This research contributes new knowledge to the design and engineering field of smart PSS.

### 2.4. Research Review

In summary, existing research on the impact of smart PSS on enterprise competitive advantage has only focused on how a component of smart PSS impacts innovation and intelligent transformation and upgrading. However, in smart PSS, intelligent technology, intelligent services, and intelligent products are interdependent and interact with each other. This narrow focus leads to an incomplete understanding of smart PSS and may result in a limited impact on competitive advantage. As an integrated whole, smart PSS offers a range of internal and external advantages, such as ecological collaboration and partner value-added benefits. However, there is no detailed description of the internal or external competitive advantages that smart PSS can provide for enterprises.

Furthermore, while existing research on the impact of smart PSS on the value creation process has discussed the significant changes brought about by digital technology, it has not thoroughly analyzed the impact of digital technology on cooperative relationships, relationship governance, and data value release in the value creation process. This may result in some enterprises being unable to effectively manage their relationships with partners in smart PSS, leading to the failure to fully realize the value of relevant data in smart PSS and ultimately hindering their digital transformation efforts. This paper adopts the widely recognized definition of smart PSS, which refers to the integration of intelligent products and electronic services using ICT technology to meet user needs. It is composed of intelligent technology, intelligent products, and intelligent services. From the perspective of the value creation process, this study aims to examine the influence mechanism of smart PSS on enterprise competitive advantage.

## 3. Research Design

The research problem addressed in this study is the question of whether the successful attainment of competitive advantages by enterprises through the development of smart PSS can be replicated, as well as the identification of potential mediating variables that connect smart PSS and competitive advantages. This concern is shared among both academic and industry communities. The objective of this study is to investigate the mediating variables and establish the logical relationship among these three variables, with the aim of unveiling the underlying influencing mechanism.

### 3.1. Research Methods

This study aims to investigate the impact mechanism of smart PSS on enterprises' competitive advantage, which belongs to the "How" type of management problems. Exploratory case studies based on grounded theory are deemed suitable for addressing such issues [35]. Currently, research on the influence mechanism of smart PSS on enterprise competitive advantage is not yet mature and mainly focuses on one aspect of its impact. Therefore, taking typical cases and adopting a root research approach can better explore and describe their interrelationships and influence mechanisms.

### 3.2. Case Selection

For this paper, Huawei Technologies Co., Ltd. (Shenzhen, China) (hereafter referred to as Huawei) is selected as the case study to examine the mechanism and the impact of Huawei's provision of intelligent connected vehicle incremental components on the enterprise's competitive advantage. This decision is motivated by several reasons: (1) Huawei is a well-known enterprise both domestically and internationally and a leader in the ICT industry. When it entered the ICV industry, it garnered considerable attention from the industry and had technological advantages that other domestic manufacturers lacked; (2) Huawei concentrates on ICT technology and provides incremental components and a complete set of intelligent solutions for intelligent connected vehicles, which aligns with the definition and composition of smart PSS; (3) Huawei collaborates with traditional automobile enterprises to achieve win–win cooperation, reflecting the current trend of Internet enterprises and traditional automobile enterprises co-building an ecosystem and establishing competitive advantages; and (4) Huawei's foray into the field of intelligent connected vehicles has ample public data that is easy to collect.

### 3.3. Data Collection

To ensure the effectiveness and credibility of the case study, this paper adopts trigonometric or polygonal measurement methods [36] in the data collection process. Specifically, it involves the following: (1) conducting semi-structured interviews with middle and senior leaders of Huawei to obtain first-hand information; (2) utilizing enterprise disclosed data, such as company press conferences, annual reports, official websites, etc.; (3) collecting relevant information published on the official WeChat official account of the enterprise or

the official account of other platforms; and (4) reviewing articles published on authoritative websites and media about Huawei's entry into the field of ICVs, as well as current hot interpretations of the ICV industry.

During the data collection process, a total of three interviews were conducted. The details of each interview are as follows: (1) The first interview occurred during the initial stage of writing. It was a face-to-face interview with five mid-level personnel from Huawei. The objective of this open-ended interview was to gain insights into Huawei's current development in relation to its entry into the smart automotive field. (2) The second interview took place after finalizing the overall framework of the writing. It was conducted via Tencent Meeting and involved semi-structured interviews with three relevant individuals. The focus of this interview was on the integration of theoretical relationship models and practical aspects within the company. (3) The third interview occurred after completing the initial draft of the paper. It was a face-to-face semi-structured interview with three individuals, specifically addressing the specific issues identified in the paper.

Each interview lasted approximately 2–3 h, and the entire data collection process spanned over a duration of seven months. The case data were collected independently by three post-graduate students from the research group between May 2022 and August 2022. Eventually, the collected data was summarized and used as research data for this paper.

## 4. Data Analysis and Theoretical Model Construction

### 4.1. Data Analysis

In terms of data types, this research primarily utilizes qualitative data. Qualitative data is descriptive, non-numerical data typically presented in textual form. In this case, the responses and viewpoints generated from open-ended questions during the interviews, as well as the textual descriptions in the secondary data, fall under the category of qualitative data. These data sources provide a deep understanding, perspectives, and subjective opinions. Therefore, the research utilizes primary data (interview data) and secondary data (Huawei reports, news articles, etc.), along with qualitative data (participant viewpoints and textual descriptions in secondary data), to support the research analysis.

Following Corbin's basic method of procedural rooting theory [37], the collected relevant raw materials were classified and analyzed based on the operational steps of rooting theory. The specific process is described as follows.

#### 4.1.1. Open Coding

Each piece of information collected about Huawei was labeled, conceptualized, and categorized. Through data analysis, summary, and induction, a total of 24 event codes, 8 concepts, and 4 main categories were extracted. Please refer to Table 1 for a detailed breakdown of the process.

**Table 1.** Open coding process.

| Case Information | Labeling | Conceptualization | Categorize |
|---|---|---|---|
| Huawei's 5G communication technology, chip technology, sensor technology, and laser radar technology are at the forefront globally. They can provide essential technical and data interaction support for intelligent connected vehicles (ICVs). | A1: Digital technology | A1: Intelligent technology | AA1: Smart PSS |
| The Huawei Hongmeng system serves as a connectivity bridge between data and equipment, enabling seamless integration and compatibility. With this system, developers can deploy multiple terminals by developing the app just once, which not only reduces operating costs but also eliminates bugs and inconsistencies. | A2: Software technology | | |

**Table 1.** *Cont.*

| Case Information | Labeling | Conceptualization | Categorize |
|---|---|---|---|
| Huawei possesses robust research and development capabilities, allowing them to provide core software and hardware components for intelligent connected vehicles (ICVs). Their expertise ensures a seamless integration of these parts with ICVs, resulting in minimal compatibility challenges. Additionally, Huawei's solutions boast low error probabilities and operating costs, further enhancing their value and reliability in the ICV ecosystem. | A3: Soft and hard matching | A1: Intelligent technology | |
| Huawei's car phone system enables the connection between the car's camera and mobile phone, enabling video calls and mobile phone screen projection in the vehicle. This integration offers convenience and flexibility in communication on-the-go. Furthermore, the system supports app installation and uninstallation, allowing users to customize their in-car experience based on their preferences and needs. | A4: Product interconnection | A2: Intelligent product | |
| Huawei HiCar utilizes 5G to enhance data interaction in the car's infotainment system. It enables seamless collection of user, product, and environmental data for better consumer insights. Through continuous analysis, HiCar adapts to changes in preferences, offering a personalized experience for car users. | A5: Data interaction | | |
| Through Huawei HiCar, users can remotely control smart home and office devices equipped with the Hongmeng system. This includes appliances like home robots, indoor purifiers, intelligent air conditioners, and related office software. HiCar provides convenience by allowing users to manage and interact with these devices from a distance, enhancing efficiency and control over their living and working environments. | A6: Remote control | | AA1: Smart PSS |
| Huawei adopts advanced laser radar solutions for autonomous driving, using three sensors. These sensors, along with the "pedestrian recognition method", enable intelligent connected vehicles to analyze and respond quickly to pedestrian-related road scenarios. This technology enhances risk detection and response, improving road safety and performance. | A7: Automatic driving service | | |
| Huawei leverages the IoT, cloud computing, and big data to offer services like data management, equipment management, operation management, terminal adaptation, collection and analysis of massive data, and more to automobile enterprises. | A8: Internet of Vehicles service | | |
| Based on big data and AI analysis, Huawei helps customers quickly build Internet of Vehicles location applications, timely and effective route planning, road forecasting and other services. | A9: High precision map cloud service | A3: Intelligent service | |
| The future development of automatic driving can be divided into two modes: single vehicle intelligence and vehicle road collaboration. Huawei's development of intelligent transportation and V2X vehicle road cloud collaboration can bring new advantages to enterprises. | A10: Car Road Collaborative Cloud Service | | |
| Through big data analysis, Huawei can predict and warn in advance of battery anomalies and spontaneous combustion. Through cloud data analysis, Huawei can formulate intelligent charging strategies to ensure battery safety and extend its service life. | A11: Three power cloud service | | |
| Huawei offers a comprehensive range of solutions for the core components of intelligent connected vehicles. It collaborates with vehicle enterprises lacking in ICT expertise to participate in vehicle manufacturing, thus achieving resource complementarity. | A12: Resource complementarity | A4: Partnership | AA2: Relationship building |

**Table 1.** *Cont.*

| Case Information | Labeling | Conceptualization | Categorize |
|---|---|---|---|
| There are two main cooperation relationships between China and automobile enterprises. The first one is deep cooperation, which entails collaboration on auto drive systems, HiCar, and on-board apps. The second one is general cooperation, which includes EIC, HiCar, and other areas in addition to the auto drive system. Huawei concentrates on ICT technology and supplies core components for intelligent connected vehicles (ICVs). By securing a significant share in the ICV market, Huawei holds a favorable position due to its expertise and contribution in this area. | A13: Partnership A14: Resource allocation | A4: Partnership | AA2: Relationship building |
| Huawei's Hongmeng system facilitates the connection of related equipment, enabling data sharing, bidirectional interaction, and acting as peripherals. This system promotes collaboration rather than one-way data transmission. | A15: Information sharing | | |
| In the context of intelligent networked vehicles, Huawei aims to provide incremental parts and establish itself as an Android-like platform in the automotive industry. By offering relevant data, software, and hardware systems, Huawei's platform enables third parties to develop their own software and enhance the overall ecosystem. | A16: Relationship Utilization | | |
| According to CITIC Securities' estimation, Huawei's provision of incremental parts for ICVs is projected to generate tens of billions of dollars in revenue over the next decade. | A17: Market benefit | A6: Benefits | AA4: Competitive advantage |
| Huawei's strategic positioning in various technology-related fields of ICVs, including laser radar, electric systems, and others, enables economies of scale and cost reduction for enterprises. This comprehensive layout allows Huawei to leverage synergies across different technologies, ultimately benefiting both the company and its partners. | A18: Operation cost | | |
| Huawei, a leading high-tech enterprise in China, has strong presence and influence in the intelligent connected vehicle (ICV) field. Collaborations with BAIC and Chongqing Xiaokang showcase its brand influence. Huawei's expertise and reputation contribute to its recognition and prominent position within the ICV ecosystem. | A19: Brand effect | A7: Sustainability | |
| Huawei can provide a full range of solutions for intelligent connected vehicles, while Baidu, Xiaomi, and other Internet enterprises still need to rely on Qualcomm's car chips. If trade decoupling is considered, Huawei's full stack advantages will be reflected. | A20: Full stack advantage | | |
| Huawei's advantages in the 5G technology field can meet the requirements of intelligent connected vehicles for high transmission speed and low delay, and provide stable and reliable services. | A21: Service stability | | |
| Huawei provides core intelligent components, enters the field of intelligent connected vehicles, and creates a multi-scene ecological chain collaboration of smart home, smart travel, sports health, video, and entertainment. | A22: Ecological chain coordination | A8: Ecological value | |
| By leveraging mobile phones and other electronic products, Huawei has formed a group of highly engaged user communities. By entering the field of intelligent connected vehicles, they aim to connect these user communities and increase user dependence. | A23: user stickiness | | |
| Automobile companies that collaborate with Huawei have experienced enhanced popularity, boosted product sales, and increased share prices. For example, the share prices of Xiaokang Shares and Huawei have both roughly doubled in just half a year. | A24: Partner value-added | | |

### 4.1.2. Axial Coding

The relationships between the 24 event codes were sorted, summarized, and clustered, and their interrelationships were analyzed. Through this process, eight subcategories and four main categories were identified, as seen in Table 2. The smart PSS category includes intelligent technology, intelligent interconnection products, and intelligent services. The relationship construction category includes cooperative relationships, while the scope of the relationship operation category includes relationship processing. Lastly, the competitive advantages of the enterprise category include efficiency, sustainability, and ecological value.

**Table 2.** Axial coding process.

| Main Category | Subcategory | Directivity | The Internal Relationship between Main Category and Sub Category |
|---|---|---|---|
| AA1: Smart PSS | A1: Intelligent technology | A1→AA1 | Intelligent technology provides technical support for smart PSS. Huawei's advantages in software, hardware, and 5G communication technology serve as the foundation for providing smart PSS for intelligent connected vehicles. |
| | A2: Intelligent product | A2→AA1 | Intelligent products act as intermediaries between smart PSS and users and enterprises. They utilize product intelligence and interconnectivity features to facilitate data interaction, remote control, and other functionalities. This allows them to obtain higher product premiums compared to competitors and establish their own unique smart PSS offerings. |
| | A3: Intelligent service | A3→AA1 | Intelligent services represent the service aspect of smart PSS, playing a significant role in enhancing user satisfaction, increasing enterprise income, and promoting the establishment of competitive advantages within the smart PSS landscape. |
| AA2: Relationship building | A4: Partnership | A4→AA2 | Partnership serves as the primary aspect of relationship building, encompassing resource and capacity complementation, as well as the establishment of partnerships. The former acts as the foundation for relationship building, while the latter represents the outcome of relationship development. Partnership plays a crucial role in influencing the enterprise's value creation process. |
| AA3: Relationship Running | A5: Relationship Processing | A5→AA3 | Relationship processing stands at the core of relationship operation, involving network resource allocation, sharing of information and knowledge, and the implementation of relationship governance mechanisms. Relationship processing directly impacts the enterprise's value acquisition and distribution process, exerting a significant influence on the value creation process of smart PSS. |
| AA4: Competitive Advantage | A6: Benefit | A6→AA4 | Benefit refers to the advantages that enterprises gain by entering the intelligent connected vehicle (ICV) market, such as higher economic benefits and lower operating costs. These advantages contribute to the formation of a competitive edge for enterprises. |
| | A7: Sustainability | A7→AA4 | Sustainability refers to the long-term and healthy development of enterprises that can be achieved by entering the intelligent network-connected automobile market. This includes factors such as brand influence, service stability, and other sustainable competitive advantages. |
| | A8: Ecological value | A8→AA4 | Ecological value, on the one hand, signifies that the enterprise has established its own leading ecosystem and formed its own ecological barriers. On the other hand, it indicates that the enterprise not only focuses on its own development but also prioritizes the development of its partners. Ultimately, this leads to the creation of unique ecological advantages for the enterprise. |

### 4.1.3. Selective Coding

Selective coding involves analyzing the logical relationship between categories using storylines. The analysis found that smart PSS, relationship construction, relationship operation, and the three main categories have a significant impact on enterprise competitive advantage. Based on this finding, a mechanism model was established—smart PSS—relationship construction—relationship operation—enterprise competitive advantage model.

### 4.2. Theoretical and Saturation Test

We conducted a theoretical saturation test on the root analysis results. If the collected data did not generate additional categories, the theory was considered to be saturated [36], and members exchanged coding materials for secondary analysis. Using the coding technology described above, we reviewed and analyzed relevant literature on smart PSS and enterprise competition. We found that Gulati, Dyer, and others divided the value creation process into three stages from the perspective of enterprise external relations: relationship construction, relationship operation, and value release [28,29]. However, when we conducted a root analysis of the case enterprise's data, we found that there were relatively few data on value release. Value release is a critical factor that directly affects a company's ability to build competitive advantage. For example, Porter et al. [3] point out that the ability to release data value has become key to affecting a company's competitive advantage.

Based on relevant literature and other business information about the enterprise, Huawei has proposed to build a "data lake" in recent years to promote data collection and governance, ultimately transforming it into providing specific services. Huawei uses smart PSS to build partnerships, collect data from users, products, and the surrounding environment, analyze and mine the value of data, and achieve value release. By expanding or improving existing businesses, Huawei establishes a unique competitive advantage. Therefore, based on our relationship mining and analysis, we construct a mechanism model of the impact of value creation on enterprise competitive advantage.

### 4.3. Model Construction of the Impact Mechanism of Smart PSS on Competitive Advantage from the Perspective of Value Creation

Huawei specializes in ICT technology and offers incremental parts and complete solutions for ICVs. The Hongmeng system allows Huawei mobile phones to connect with Huawei-related devices like smart home systems, smart sports equipment, and more, transferring a large number of user markets accumulated in the mobile terminal. Furthermore, suppliers of ICV incremental parts have extended the industrial value chain by finding matching application scenarios for Huawei's 5G technology, communication facilities, etc. This has reduced the risk of participating in the manufacturing of finished vehicles and improved enterprises' competitiveness in the new environment.

Through data collection, analysis, and induction of Huawei's relevant data, we analyzed information from various channels, obtained relevant concepts from the case data, and established a logical relationship between them. During the research process, we thoroughly studied smart PSS, value creation literature, and Huawei's related businesses, including their mobile phone business, Hongmeng ecosystem, and data lake construction. Based on this analysis, we clarify the logical relationship between categories and reveal the internal mechanism of smart PSS's impact on competitive advantage from the perspective of value creation, as shown in Figure 1.

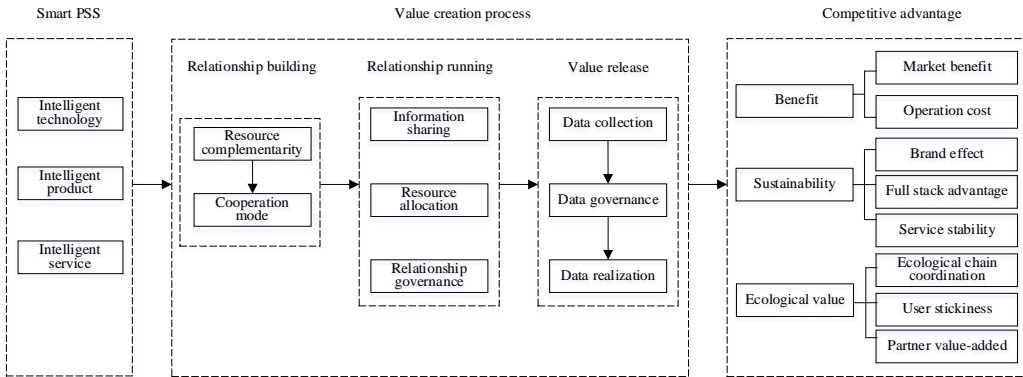

**Figure 1.** Mechanism model of the impact of the smart PSS value creation process on competitive advantage.

## 5. Model Interpretation and Research Findings

This study has established a conceptual model of the internal mechanism of the impact of smart PSS on competitive advantage from the perspective of value creation based on the analysis of Huawei's value-added parts of ICVs. To further clarify the relationship between model concepts and the process of value creation, this research starts with the composition of smart PSS, which includes intelligent products, intelligent technology, and intelligent services. This study also examines the process of value creation, including the impact of relationship construction, relationship operation, and value release on the competitive advantage of enterprises, to explain its internal mechanism process.

### 5.1. Impact of Smart PSS Composition on Enterprise Competitive Advantage

Huawei is focusing on ICT technology to help automobile enterprises build high-quality cars by providing intelligent connected vehicles with incremental parts and a complete set of intelligent connected vehicle solutions, known as smart PSS. This allows Huawei to extend the industrial chain around its main business. According to CITIC Securities' estimation, Tier 1 suppliers are expected to generate tens of billions of dollars in market revenue for Huawei in the next ten years. By providing smart PSS to traditional car enterprises, Huawei has attracted a large number of users, improved brand awareness and stock prices of traditional car enterprises, and promoted their development. For instance, after Chongqing Xiaokang cooperated with Huawei, their brand awareness significantly improved, and their stock price doubled in half a year. Currently, Huawei has established cooperative relationships with many traditional car enterprises like Chang'an, BAIC, and Xiaokang around smart PSS. By integrating resources from all parties, this cooperation allows for better business collaboration, reduces enterprise operating costs, and achieves scale effects. This is of great significance for Huawei in establishing a Hongmeng ecosystem.

Smart PSS is supported by intelligent technology, which Huawei excels in due to its unique technical advantages and mature R&D system in 5G, big data, chips, and other fields. Leveraging these advantages, Huawei can meet the requirements of intelligent connected vehicles for low latency and high-quality data transmission and information interaction, providing an excellent scenario for smart PSS technology applications. As one of the few internet enterprises that have entered the field of intelligent connected vehicles, Huawei is the only enterprise capable of providing a complete set of solutions for intelligent connected vehicles through cooperation with traditional vehicle enterprises, which is smart PSS. Moreover, intelligent technology serves as a foundation for enterprises to play a leading role and build an ecosystem. Enterprises can utilize intelligent technology to achieve collaboration between internal businesses, closely linking the upstream and downstream industrial chains, promoting the formation of an enterprise-centered ecosystem, and establishing enterprise competitive advantages.

Huawei has formed smart PSS, which includes intelligent cockpit, intelligent driving, intelligent electric, intelligent networking, and intelligent car cloud. This enables enterprises

to provide a full range of intelligent products, technologies, and services. In the face of potential trade decoupling risks, Huawei's advantage of "full stack service" is highlighted, facilitating all-around business cooperation with traditional car enterprises and achieving a win–win cooperation between the two sides. By utilizing intelligent products, enterprises can collect relevant data about products, users, and the surrounding environment to provide personalized, high-quality solutions, achieve a unique premium beyond the product itself, and enhance user dependence. For instance, smart products like Hicar, an intelligent connected car, can be connected with mobile phones, smart homes, smart offices, and other devices through the Hongmeng system, leading large numbers of mobile customers into the field of intelligent connected cars. This helps enterprises integrate businesses related to the industrial chain, achieve ecological coordination, and establish competitive advantages.

In April 2023, Huawei showcased a vehicle at the Shanghai Auto Show that comes equipped with the latest Huawei ADS 2.0 advanced intelligent driving solution. This vehicle stands out as it offers intelligent driving assistance without the need for high-definition maps. Instead, it relies solely on visual perception and fusion sensing technologies. It is anticipated that this vehicle will be introduced as the Wanjie M5 Advanced Intelligent Driving Edition, given previous announcements made by the Ministry of Industry and Information Technology. With the combined capabilities of ADS 2.0 and HarmonyOS 3.0, this new vehicle is set to establish a higher level of competitiveness within the smart electric vehicle market. It underscores Huawei's commitment to innovation by delivering cutting-edge solutions in the automotive industry. The integration of advanced intelligent driving features and a robust operating system holds great promise for the future of autonomous vehicles.

Based on the above research and analysis, we propose the following research propositions:

**Proposition 1.** *The provision of smart PSS by enterprises has a positive impact on the establishment of competitive advantages in terms of enterprise benefits, sustainability, and ecological value.*

**Proposition 1a.** *Within smart PSS, intelligent technology positively impacts the establishment of competitive advantages in terms of enterprise service stability, ecological coordination, and reducing operating costs.*

**Proposition 1b.** *Within smart PSS, intelligent products and services have a positive impact on enterprises to establish competitive advantages in terms of market efficiency, full stack advantage, user stickiness, and partner value-added.*

*5.2. Impact of Smart PSS on the Value Creation Process*

During the process of analyzing case data, it was found that within the smart PSS environment, the widespread embedding of intelligent technology in products and services changed the relationship, subject, resources, and key elements of the enterprise value creation process [27]. With complementary resources, enterprises can utilize smart PSS and partners to conduct in-depth or general cooperation, establish cooperative relationships, expand the scope of resources for enterprise value creation, and facilitate characteristics of multi-party participation in value creation. To focus on the rational allocation of internal system resources of smart PSS and enhance customer experience, enterprises share information about customers, products, and the surrounding environment with partners, form a series of rules and governance mechanisms within smart PSS to ensure the operation of the system, and leverage the support of intelligent technology and the wide use of intelligent connected products to greatly enhance their ability to collect, analyze, and use data, leading to the development of more intelligent services. In the smart PSS environment, data has become the core resource for value creation. Therefore, the digitalization of the enterprise value creation process is evident, the influence of partnerships is significant, and the value of data is prominent.

Enterprises leverage smart PSS to establish cooperation platforms and expand the scope of resources for value creation. Within the system, enterprises establish cooperation based on business contacts and resource complementation. Smart PSS stakeholders, with a focus on resource allocation and meeting customer requirements, share information about products, users, and the surrounding environment within smart PSS. Relevant service providers can develop various services on the basis of the platform after forming the internal relationship rules of the system. Case data shows that Huawei provides in-depth and general cooperation with traditional car enterprises through smart PSS by integrating the resources of internet and manufacturing enterprises, building cooperative relationships, and enabling customers, suppliers, stakeholders, and other parties to participate in value creation. Huawei shares data and information with internal members of the system to configure internal resources of the system more reasonably and improve customer experience.

Huawei has expressed that the Smart Selection Car Mode has been developed on its solid foundation, offering a complete range of technologies and solutions. Moreover, it incorporates Huawei's extensive experience of over a decade in aspects related to consumer engagement, including customer experience design, product marketing, brand marketing, user experience design, and expertise in quality assurance. Huawei emphasizes the importance of close collaboration with automotive manufacturers to jointly create exceptional vehicles. By leveraging its comprehensive capabilities, Huawei aims to empower these manufacturers and enhance the overall quality and performance of the vehicles. This collaborative approach ensures that the best possible products are delivered to customers, aligning with Huawei's commitment to excellence and customer satisfaction. The case data highlights that smart PSS has accumulated a significant amount of data about products, users, and the external environment. Various service providers on the platform analyze changes in this data to understand the product usage status and potential user needs. They then provide specific solutions or services for customers through data realization to realize the value of the data. For example, although Huawei has collected a lot of relevant data while providing smart PSS for ICVs, it is difficult to fully harness the value of the data and convert it into services solely relying on Huawei. Rather, smart PSS plays a more "platform" role in this regard.

Huawei has made significant strides in the automotive industry by introducing a range of electronic products and solutions. In addition to their involvement in battery technology, Huawei has also delved into the realm of "Three Electric" solutions, with a particular emphasis on what they refer to as the "Four Intelligences": intelligent cockpit, intelligent driving, intelligent connectivity, and intelligent vehicle control. By focusing on these four key areas, Huawei has established a comprehensive automotive ecosystem. This ecosystem encompasses various aspects of automotive technology, aiming to enhance the intelligence and connectivity within vehicles. With its expertise and innovation, Huawei is actively contributing to the advancement of the automotive industry and creating an environment where smart and connected vehicles can thrive. Therefore, smart PSS expands industry boundaries and brings all stakeholders to the smart PSS platform where they can tap into the resources they need and release value around the data. The platform participants take what they need and leverage the data to create value.

Based on the research and analysis provided, the following research propositions can be established:

**Proposition 2.** *Smart PSS has had a positive impact on the value creation process by transforming the scope, logic, and resources involved.*

**Proposition 2a.** *Smart PSS has changed the scope and logic of the value creation process. The platform's characteristic of multi-party participation in the value creation process is evident as it has moved from a product-oriented approach to a service-oriented ecosystem.*

**Proposition 2b.** *Smart PSS has also transformed the resources utilized in the value creation process from traditional element resources to data resources. Data has become a key resource in the smart PSS environment and is essential for creating value within the platform.*

*5.3. Impact of the Value Creation Process on Enterprise Competitive Advantage*

Data analysis shows that the value creation process of smart PSS plays a crucial role in understanding the impact of smart PSS on the competitive advantage of enterprises. Huawei has entered the intelligent connected vehicle field and provided smart PSS, with its value creation process significantly related to the successful establishment of competitive advantages regarding efficiency, sustainability, and ecological value. Huawei and traditional car companies have established deep or general cooperation based on whether to provide an auto drive system. The position of enterprises in relationship operation, and the ability to release data value under different cooperation relationships, emphasizes different aspects of building enterprise competitive advantage, as shown in Table 3.

**Table 3.** Differences between in-depth cooperation and general cooperation.

| | | Deep Cooperation | General Cooperation |
|---|---|---|---|
| Relationship structure | Cooperation mode | Enterprises provide the majority of core components for intelligent connected vehicles (ICVs), including the autonomous driving system. | The focus is on module technology output and establishing electronic collaborations with vehicle enterprises in the non-autonomous driving field, such as three electric, wireless vehicle modules, HiCar, etc. |
| | Complementary resources | This is applicable to traditional automobile enterprises with weaker foundations as it enables them to expedite their entry into the intelligent connected automobile industry. | This is applicable to traditional car enterprises with strong strength who do not want relevant data to be mastered by other enterprises. |
| Relationship operation | Information sharing | As the primary information provider in smart PSS, enterprises share relevant information with their partners. | Enterprises share relevant information as participants through relevant information provided by core enterprises. |
| | Resource allocation | Enterprises hold a central position within smart PSS and possess control over the internal resource allocation of the system. | Enterprises are dominant in smart PSS and subject to the internal resource allocation of the system. |
| | Relationship Governance | The enterprise takes the lead in establishing the internal operational rules and benefit distribution mechanisms of the system. | The enterprise obeys the operation rules and benefit distribution mechanism established by the core enterprise. |
| Value release | Data collection | Through the core smart PSS, enterprises gather and analyze significant amounts of data related to users, products, and the surrounding environment. | The enterprise simply provides technology, does not collect and master data, and the data belongs to traditional automobile enterprises or other core enterprises. |
| | Data governance | The enterprise possesses the authority to manage, utilize, and distribute data assets, showcasing strong governance capabilities. | The enterprise loses the ownership, management right, use right, and data governance ability of data. |
| | Data realization | The enterprise possesses abundant data resources and high data fluidity, which is conducive to fully unlocking the value of data. | Enterprises have less data and poor data liquidity, which is not conducive to the release of data value. |
| Competitive edge | | Enterprises prioritize ecosystem value and strive to establish ecological advantages. | The focus is on the advantages of achieving benefits and sustainable development. |

When traditional car enterprises with weak foundations are eager to enter the field of intelligent cars, they establish deep cooperation with enterprises that provide most of the core parts of intelligent connected cars based on resource complementation. The enterprise is in a dominant position in the relationship operation, sharing relevant information with partners, mastering the dominant power of resource allocation, and establishing relevant benefit distribution mechanisms. Data collection is carried out through intelligent products, and data governance and realization are carried out through the smart PSS platform, which benefits enterprises and partners and forms an ecosystem dominated by enterprises, participated in by many parties, and characterized by win–win cooperation. When traditional vehicle enterprises have strong capabilities and do not want their vehicle data to be controlled by other enterprises, they will build general partnerships with traditional vehicle enterprises focused on module technology output. They conduct electronic cooperation with traditional vehicle enterprises in the field of non-automatic driving, including the three electricity, wireless onboard modules, HiCar, etc. Under this partnership, the enterprise is in a dominant position in the relationship operation and the ownership of data belongs to the traditional car enterprises or other core enterprises. The value release capacity is low, and more benefits come from module technology output.

The collaboration between Huawei and Cyrus represents a new and innovative approach to cross-industry integration. It signifies the merging of ICT companies and automakers, creating a practical example of deep integration between the digital economy and the physical economy. As the automotive industry rapidly progresses towards intelligent development, the concept of "going it alone" is no longer the dominant theme for future growth. Instead, the industry's future lies in the inevitable direction of "cross-industry integration". This integration involves the convergence of automotive, energy, transportation, and artificial intelligence sectors. Huawei recognizes this direction and aims to provide comprehensive empowerment throughout the entire value chain. By doing so, they contribute to the success of Wanjie in becoming the best intelligent connected vehicle. Through their expertise and support, Huawei helps drive the integration of intelligence and connectivity in the automotive industry, shaping the future of smart transportation.

Based on the above research and analysis, we propose the following research propositions:

**Proposition 3.** *The value creation process of smart PSS promotes competitive advantages in terms of efficiency, sustainability, and ecological value.*

**Proposition 3a.** *In the value creation process of smart PSS, deep cooperation means that enterprises account for a large proportion of the core part of intelligent connected vehicles. This will help enterprises establish competitive advantages in terms of full-stack advantage, market efficiency, user stickiness, brand effect, partner value-added, service stability, and ecological chain collaboration.*

**Proposition 3b.** *In the process of value creation of smart PSS, general cooperation, that is, the proportion of enterprises in the core part of ICVs is relatively small, will help enterprises establish competitive advantages in market efficiency, brand effect, and partner value-added.*

*5.4. Intermediary Role of the Value Creation Process*

After analyzing case data and relevant literature, it has been found that smart PSS enterprises have a significant influence on the formation of competitive advantage. However, simply using smart PSS composition is not enough for an enterprise to establish a unique and sustainable competitive advantage. The key lies in clarifying the internal mechanism of the value creation process of smart PSS to continuously create value for the enterprise.

For instance, Huawei entered the field of intelligent connected vehicles leveraging its advantages in information and communication technology (mainly represented by 5G technology, Hongmeng ecosystem, etc.), which could provide various intelligent products and services to form a smart PSS. Through a complementary resource relationship with traditional vehicle enterprises, they provided smart PSS to build cooperation and establish a

relationship that includes providing core components of intelligent networked vehicles. The collaboration between Huawei and Cyrus has proven to be successful in leveraging each other's strengths, leading to continuous advancements in technological accumulation and delivering high-end product experiences. This partnership has propelled the AITO Wanjie brand to become a rising star in the domestic automotive market. In the J.D. Power China New Energy Vehicle Initial Quality Study (NEV-IQS), the Wanjie M5 garnered recognition by ranking first in the mainstream plug-in hybrid segment for its quality performance. This achievement has gained both consumer and authoritative institution recognition. The quality advantage of the Wanjie M5 can be attributed to various factors, with Huawei's deep empowerment playing a crucial role. Huawei's system support has seamlessly integrated technology and quality standards into the research and development, production, and delivery stages of the Wanjie M5. This integration has significantly enhanced its quality foundation and bestowed leading software and hardware capabilities, as evidenced by multiple industry evaluations. Huawei's empowerment is evident in several aspects of the Wanjie M5. For instance, the Wanjie M5 Intelligent Driving Edition features a million-dollar luxury car-level all-aluminum chassis, providing exceptional handling. The innovative intelligent range extension technology ensures constant power and ultra-low energy consumption. The HarmonyOS 3.0 smart cockpit and HUAWEI ADS 2.0 advanced intelligent driving solution create an intelligent driving experience. In an impressive achievement, the AITO Wanjie brand reached the milestone of producing its 100,000th vehicle within just 15 months, making it the "fastest new energy vehicle brand to achieve this milestone". This milestone further emphasizes the success and rapid growth of the AITO Wanjie brand in the market.

Once the enterprise and traditional automobile enterprise established a partnership, the position of both parties in resource allocation, information sharing, and relationship governance was determined according to the proportion of smart PSS provided by the enterprise in the core parts of intelligent networked vehicles. By operating the partnership formed by the smart PSS, data collection, governance, and realization were carried out to release the value of the data. In this way, the market efficiency of enterprises can be increased, the industrial chain can be extended, and the ecological chain can be synergistically enhanced to help enterprises establish competitive advantages, which includes building enterprise smart PSS benefit advantages, smart PSS sustainability advantages, and smart PSS ecosystem advantages.

Based on the above analysis, we propose the following research proposition:

**Proposition 4.** *The value creation process of smart PSS, including relationship construction, relationship operation, and value release, plays an intermediary role in establishing smart PSS and enterprise competitive advantage.*

Overall, smart PSS has a positive impact on the competitive advantages of enterprises, benefiting their profitability, sustainability, and ecological value. It achieves this through the implementation of intelligent technologies that ensure service stability, ecological coordination, and cost reduction, as well as the provision of intelligent products and services that enhance market efficiency, offer full-stack advantages, increase user stickiness, and provide added value to partners. These factors collectively contribute to the competitive advantages of enterprises.

Smart PSS brings about changes in the scope, logic, and resources involved in the value creation process, thereby positively influencing it. It shifts the focus from product-oriented approaches to service-oriented ecosystems and transforms traditional elemental resources into data resources. This transformation highlights the importance of multi-party participation and data as crucial resources for value creation within the smart PSS environment.

The value creation process of smart PSS facilitates the development of competitive advantages in terms of efficiency, sustainability, and ecological value. Through deep collaboration, enterprises can secure a significant share in the core aspects of intelligent connected

vehicles. This allows them to establish competitive advantages in terms of full-stack capabilities, market efficiency, user stickiness, brand effect, partner value-added services, service stability, and ecosystem synergy. However, in cases of general collaboration, where enterprises have a smaller proportion in the core aspects of intelligent connected vehicles, they can still establish competitive advantages in terms of market efficiency, brand effect, and partner value-added services.

Lastly, the value creation process of smart PSS acts as an intermediary in establishing both smart PSS and the competitive advantages of enterprises. It involves relationship building, relationship operation, and value release. These three aspects interact with each other, collectively driving the development of smart PSS and forming competitive advantages for enterprises.

*5.5. Case Expansion Analysis*

Further research selects Shaanxi Automobile Group (Xi'an, China), Shaangu Group (Xi'an, China), and other companies that develop or provide intelligent PSS as research subjects, comparing them with Huawei. These three case companies are prominent enterprises in their respective industries and come from different sectors, demonstrating industry diversification. They include Shaanxi Automobile in the field of commercial vehicle Internet of Things and Shaangu in the field of large-scale equipment manufacturing. These three companies have many years of experience in developing or implementing smart PSS, and there are some differences in terms of industry, characteristics, and market, as shown in Table 4.

**Table 4.** Horizontal comparison of case company characteristics.

| Company | Industry Characteristics | Market Objectives | Demand Characteristics | Product Features |
|---|---|---|---|---|
| Shaanxi Automobile | Heavy-duty truck industry: Relying on Shaanxi Automobile Group's main engine factory, significant policy advantages and tapping into the value of product data | Focusing on products with a customer-centric approach | Rapid demand growth; Concentrated demand; Low level of customization | Fast iteration and updates; Multiple functions; High technological content; High complexity |
| Shaangu | Energy equipment: Strong awareness of transformation, a strong organizational culture, and leading technology in the industry. | Leading customer demand through comprehensive solution offerings | Slow demand growth; Concentrated demand; High level of service customization. | Slow product iteration and updates; Fast service innovation; High technological content and complexity |
| Huawei | Information and Communication: Leading technology, strong research and development capabilities, and a strong brand influence | Technological innovation; Leading market customers | Rapid demand growth; Dispersed demand; High degree of product customization. | Software and hardware integration; Fast iteration and updates; Diverse functions; High technological content; High complexity. |

Shaanxi Automobile, Shaangu, and Huawei have significant differences in terms of enterprise characteristics, market objectives, customer demands, and product features. These differences lead to the adoption of differentiated service strategies by each company based on their industry characteristics, aiming to achieve their predetermined goals. These differences are reflected in the process of resource integration, dynamic capabilities, and the construction of competitive advantages for the case companies. Shaanxi Automobile focuses on solving customer problems through product concentration. Its market environment is primarily influenced by two factors: product-based customer demands and government regulations. To respond to the market environment, the company needs to leverage its products, explore the value of data utilization, provide diversified and personalized services to users, and utilize policy advantages to collaborate with relevant

departments. This enables them to transform their manufacturing advantages into special service advantages, continuously extending the company's "service depth" around their products. On the other hand, Shaangu solves market customer problems through product services. Their products, large-scale energy equipment, have a lower level of customization, but the level of service customization is higher. The company develops product-based solutions and solutions based on the customer supply chain. During their development process, they not only strive to improve product quality and enhance product competitiveness but also focus on expanding product-based services. They conduct in-depth analyses of customer production capacity, energy consumption, finances, and other related aspects to provide corresponding services, promote the development of partners, and form a strategic approach of "service depth" based on their products and "service breadth" based on the supply chain upstream and downstream. Overall, each company adopts different strategies based on their industry characteristics to cater to their market objectives and meet customer demands.

Through the comparison of cases, it is found that Shaanxi Automobile and Shaangu have achieved rapid growth in sales revenue and profit margin, as well as an increase in the proportion of service revenue, by promoting dynamic capabilities through resource integration. This is consistent with the research results that show how service-oriented companies can enhance their profitability. As for Huawei's Intelligent PSS, which is still in the early investment stage, its resource integration effects cannot be precisely measured by financial indicators at present. However, the research reveals that Huawei has prominent advantages in terms of product service quality, risk management, and cost control. Furthermore, the analysis indicates that there are certain differences in the outstanding aspects of competition advantages in service-oriented companies. Shaanxi Automobile currently focuses more on efficiency advantages and then transitions to gain sustained development advantages, while the impact of ecological value advantages is relatively small. Shaangu significantly improves both the profitability and sustainability advantages through transformation, bringing about strong ecological value advantages. Huawei, on the other hand, enhances its ecological value advantages significantly by diversifying into cross-industry transformation and expanding the HarmonyOS ecosystem. Business transformation is promoted to ensure sustained development, and leveraging ecological and technological advantages may bring substantial operational benefits to the company.

Multiple case studies have found that compared to Shaanxi Automobile and Huawei, Shaangu's sustained advantages are more prominent. The reason is that Shaangu not only provides product solutions in the process of integrating service-oriented resources but also utilizes the "Chain Easy Access" platform to provide comprehensive service plans for customer development, expanding service depth and width around products and supply chains. At the same time, Shaangu has utilized resources from various sources such as the government and strategic partners to establish a comprehensive market control system, combined with the intelligence of products and services, significantly enhancing the enterprise's ability to perceive and capture the market. Shaanxi Automobile focuses on the Internet of Vehicles, develops product-based big data analysis, intelligent management of waste trucks, and other services, expands the depth of enterprise services, increases the proportion of service revenue, and has a significant impact on enterprise efficiency. Huawei provides intelligent vehicle solutions around its main business, cooperates with traditional car companies, and enters the intelligent vehicle market as a supplier. The automotive industry chain is essentially still dominated by traditional car companies. Unlike Shaanxi Automobile and Shaangu, Huawei provides core components for smart cars, integrating them into the Hongmeng ecosystem, achieving cross-industry ecological collaboration, bringing significant ecological value, and further expanding the service scope. Therefore, for Huawei, utilizing intelligent PSS for service-oriented transformation through resource integration mainly brings ecological value advantages. The next step is to use ecological value advantages to deeply expand services and bring high-efficiency advantages.

In short, firstly, intelligent PSS enterprises need to continuously develop and improve the types of services through data analysis, adjust their organizational structure to adapt to market product changes, improve market response speed, and increase market initiative. Then, intelligent PSS enterprises adopt a diversified profit model, where big data can not only analyze products but also users. Through data analysis, they can develop more related businesses, enhance collaboration between businesses, and reduce operational risks. Finally, intelligent PSS enterprises should be user-centric, with the aim of improving user satisfaction, achieving mutual benefit and win–win situations among multiple parties, and promoting common development.

## 6. Conclusions and Enlightenment

One of the prominent features of the development of intelligent product service systems (PSS) is the application of smart technologies in the context of products and services, which has garnered widespread attention due to its impact on changes in products and services. Many studies have discussed how to design intelligent PSS to achieve sustainable development for companies and have started exploring the relationship between the application of smart technologies and value creation and competitive advantage, but the costs associated with developing smart technologies, including the overall life cycle costs and opportunity costs brought about by the design, manufacturing, operation, and maintenance of products and services, have been largely overlooked. This has led to companies pursuing technological progress in a one-sided manner, investing excessively, and facing risks such as financial strain and bankruptcy.

One of the innovations of this article lies in establishing the correlation between the specific objectives of building competitive advantages and the basic components of intelligent PSS and using a result-oriented approach to integrate intelligent products, services, and technologies. It particularly explores the theoretical contributions of the development of smart technologies to stakeholders' interests, including suppliers, customers, and technology service providers, thereby complementing existing research that is limited to discussing the application of smart technologies within companies.

This paper employs the research method of single case root analysis and examines the mechanism of smart PSS composition, which includes intelligent products, intelligent technology, and intelligent services, from the perspective of value creation. This study aims to explore how these elements impact the competitive advantage of enterprises and analyze their unique intermediary role in the value creation process of smart PSS. The main conclusions of this research are as follows:

(1) Smart technologies play a crucial role in smart PSS. By leveraging technologies such as artificial intelligence, big data, the Internet of Things, and cloud computing, smart products gain self-awareness, self-monitoring, self-control, and self-decision-making capabilities. These smart technologies enable products to automatically collect and analyze relevant data, make adjustments and decisions based on the data, and provide personalized and high-quality services. Moreover, smart technologies support businesses in gaining a competitive advantage by expanding the value chain. Through collaboration with other companies or partners, smart products can achieve interoperability with other devices and services, forming a complete smart ecosystem. For example, smart cars can collaborate with intelligent traffic systems, charging stations, and navigation systems to offer more intelligent and convenient transportation solutions. Additionally, smart technologies bring economic benefits and reduce operating costs for businesses. By employing automated and intelligent production processes, companies can improve efficiency and minimize labor and resource costs. Simultaneously, smart products can collect and analyze user usage data, enabling businesses to better understand user needs and market trends, optimize product design and service strategies, and enhance competitiveness and profitability.

(2) In the smart PSS environment, the core resources for value creation have shifted from products to the associated data provided by products. This shift has led to a

data-centric approach in relationship building, operation, and value release, offering a new pathway for enterprises to establish competitive advantages through data utilization. Multi-party involvement is a crucial characteristic of value creation in smart PSS, expanding the boundaries of value creation. Enterprises focus on both internal and external interests, cultivating stable and sustainable partnerships. Smart PSS automatically collects real-time data on products, users, and the environment. Resource allocation and information sharing revolve around the effective management and utilization of this data to generate more value. Those who have control over the data wield influence in relationship operations, laying the foundation for competitive advantages. Therefore, in the value release stage, enterprises and relevant partners concentrate on fully leveraging data through monetization to meet the needs of all parties, providing differentiated services and establishing competitive advantages for the enterprise.

As smart PSS continues to evolve, enterprises are increasingly emphasizing data collection, analysis, and application. By analyzing data such as user behavior and preferences, enterprises can gain a better understanding of user needs, accurately position products and services, and provide personalized recommendations, thereby enhancing user satisfaction and loyalty. Simultaneously, by monitoring and analyzing product and system data, enterprises can predict faults and optimize maintenance, improving product reliability and stability.

Data exchange and sharing play a pivotal role in smart PSS. Enterprises can share data with suppliers, partners, and other relevant parties to achieve more efficient collaboration and resource integration. For instance, intelligent manufacturing companies can share real-time production data with component suppliers to optimize production plans and achieve lean management in the supply chain. This data-driven collaboration model substantially enhances the efficiency and flexibility of the entire value chain, providing significant competitive advantages to the enterprise.

(3) In the smart PSS environment, companies establish various modes of cooperation with complementary resource partners, which have different effects on their competitive advantages. These effects are influenced by factors such as technological differences, mutual trust, and dependence. There are primarily two modes of cooperation: deep cooperation and general cooperation.

In the deep cooperation mode, companies establish close relationships with their partners and play a significant role in the value chain of the products. In this mode, companies have control over most of the core values of the products and the relevant data. This enables companies to better understand user needs and provide more personalized services, thereby enhancing user loyalty. Furthermore, deep cooperation improves collaboration efficiency among partners and enhances overall service stability through ecosystem synergy. Ultimately, this helps companies establish a unique ecosystem advantage, enabling them to stand out in market competition.

On the other hand, partnerships in the general cooperation mode are relatively less intensive. Cooperation between companies and partners primarily focuses on specific areas or segments without the same level of comprehensive coordination as in deep cooperation. In this mode, cooperation between companies and partners is primarily based on shared interests with a lower degree of mutual dependence. Although partnerships in the general cooperation mode may lack depth and breadth, they can still provide specific resources and capabilities to meet specific market demands.

The research findings of this article are consistent with the conclusions of Configuration Three and Configuration Four in the study of product service system evolution (Gaiardelli et al., 2021) [38]. Through a single case study of a supplier in the new energy vehicle industry, the transition from a product service system to an intelligent product service system is verified as a new change.

The choice of a single case study in this article is mainly aimed at analyzing the reasons behind Huawei's repeated emphasis on its non-car manufacturing declaration, gaining a comprehensive understanding of the issues of concern in the new energy vehicle industry, verifying the development of intelligent PSS, and deriving new theoretical perspectives from a specific case.

Although this article provides some preliminary exploration and analysis, obtaining related research conclusions by excavating case enterprise data and relevant literature, as an exploratory case study, it still has some defects. This study has several limitations that should be acknowledged. Firstly, the reliance on a single case analysis may restrict the generalizability of the findings. It would be beneficial to include a larger and more diverse sample size, encompassing multiple cases from various industries, to enhance the external validity of the research. Secondly, this study is constrained by limited access to primary data from within the organization. This restricted access may have hindered the ability to obtain a comprehensive understanding of the factors influencing competitive advantage in the intelligent PSS environment. Future studies should strive to establish stronger relationships with organizations to gain better access to primary data. Furthermore, the absence of specific indicators or measures to evaluate competitive advantage is another limitation of this study. Incorporating robust metrics and performance indicators would provide a more rigorous assessment of the impact of smart PSS on competitive advantage. Moreover, the study primarily focuses on the direct impact of smart PSS on competitive advantage, overlooking its long-term effects and sustainability. To comprehensively explore the dynamic nature of competitive advantage, future research should consider examining the long-term implications and sustainability of implementing smart PSS.

To address these limitations, future studies can adopt a comparative approach by analyzing companies that have implemented smart PSS and those adhering to traditional business models. This type of comparative analysis would facilitate a deeper understanding of the distinct advantages and challenges associated with smart PSS. Additionally, it could offer valuable insights and strategies for traditional firms to effectively leverage intelligent technologies.

**Author Contributions:** Conceptualization, L.H.; Methodology, Y.Y.; Formal analysis, J.S. All authors have read and agreed to the published version of the manuscript.

**Funding:** This research was funded by the National Social Science Foundation of China, grant number 20BGL018.

**Institutional Review Board Statement:** Not applicable.

**Informed Consent Statement:** Not applicable.

**Data Availability Statement:** No new data were created or analyzed in this study. Data sharing was not applicable to this study.

**Acknowledgments:** The authors appreciate the anonymous reviewers for their constructive comments and suggestions that significantly improved the quality of this manuscript.

**Conflicts of Interest:** The authors declare no conflict of interest.

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
