# Peer review of "Exploring the Influence of Smart Product Service Systems on Enterprise Competitive Advantage from the Perspective of Value Creation"

_sustainability, doi:10.3390/su151813828_

Round 1
Reviewer 1 Report
Dear authors,
the paper presented to Sustainability with the main focus on PSS has no originality, and a very low contribution to the scholarship.
It has very low strenght of evidence. low scientific soundness.
The article has used only the view of one company with no possibility to test the existant empirical evidence of other aurthors. Controll experiments are not possible.
The research design is of very poor quality resulting in poor arguments and results.
In the best scenario it could be a case study, but then the research design, questions, hypotheses and methods should be clearly stated and explained.
Further the conclusions thoroughly supported by the results presented in the article or referenced comparing the findings with other secondary literature.
best regards
the reviewer
Reviewer 2 Report
This article clearly demonstrates the role of Smart PSS in enterprise competitive advantage using the case of Huawei Technologies. The discussion in the article is well explained according to the theory used. Why Huawei and what the research gap is—or at least the difference between the current article and the previous articles—have been explored. However, some more explanation of data collection needs to be given, such as the time frame to collect the data with three postgraduate students. The authors may provide this information in the data collection section.
Reviewer 3 Report
A well-written paper, thank you to the authors. Following are a few suggestions for further improvement:
1) Please revise the title of this paper. Instead of the word 'research' please use 'explore' or any other relevant word for making the title more sound. Also please use the full form of PSS in the title.
2) The introduction part is well-written. Please add the significance of the study at the end of the Introduction section, because the audience must know the importance of your research.
3) In the methodology section, please discuss the research data in more detail such as primary or secondary data or both; qualitative data or quantitative data or both types of data?
4) Also, please discuss in detail the semistructured interview. Such as, How many interviews were conducted; interview duration; detail of interview protocol and interview conducting process etc.
5) Please discuss the study limitation and scope for future studies more elaborately. Presently it's very precise.
6) Please add more recent citations as this paper used only 23 references.
Thank you and best wishes.
Reviewer 4 Report
The manuscript has some merit for consideration, but needs revision before consideration. The following points is to be considered while revising the paper:
1. What is the innovation present in the paper and should be highlighted.
2. Please expand Smart PSS while it appeared first time in the abstract.
3. Competitive advantage and value creation are addressed, but hoe it leads to the sustainability.
4. Sustainability related to the present work must be elaborated.
5. No numerical or graphical representation of results is presented, why?
6. Compare the results without Smart PSS and with Smart PSS.
7. More analysis is needed in discussion.
8. Improve the conclusion.
9. English needs further improvement.
English needs further improvement.
Round 2
Reviewer 1 Report
Dear authors,
the paper has still no originality, and a very low contribution to the scholarship.
It has very low evidence therefore it should be revised and sent to another journal.
best regards,
the reviewer,
Author Response
Thank you for your work.
Point : The paper has still no originality, and a very low contribution to the scholarship. It has very low evidence therefore it should be revised and sent to another journal.
Response :
One of the prominent features of the development of intelligent Product-Service Systems (PSS) is the application of smart technologies in the context of products and services, which has garnered widespread attention due to its impact on changes in products and services. Many studies have discussed how to design intelligent PSS to achieve sustainable development for companies and have started exploring the relationship between the application of smart technologies and value creation and competitive advantage, but the costs associated with developing smart technologies, including the overall life cycle costs and opportunity costs brought about by the design, manufacturing, operation, and maintenance of products and services, have been largely overlooked. This has led to companies pursuing technological progress in a one-sided manner, investing excessively, and facing risks such as financial strain and bankruptcy.
One of the innovations of this article lies in establishing the correlation between the specific objectives of building competitive advantages and the basic components of intelligent PSS, and using a result-oriented approach to integrate intelligent products, services, and technologies. It particularly explores the theoretical contributions of the development of smart technologies to stakeholders' interests, including suppliers, customers, and technology service providers, thereby complementing existing research that is limited to discussing the application of smart technologies within companies.
The research findings of this article are consistent with the conclusions of Configuration Three and Configuration Four in the study of Product-Service System evolution (Gaiardelli et al., 2021). Through a single case study of a supplier in the new energy vehicle industry, the transition from a Product-Service System to an intelligent Product-Service System is verified as a new change.
The choice of a single case study in this article is mainly aimed at analyzing the reasons behind Huawei's repeated emphasis on its non-car manufacturing declaration, gaining a comprehensive understanding of the issues of concern in the new energy vehicle industry, verifying the development of intelligent PSS, and deriving new theoretical perspectives from a specific case.
I have included these contents in the conclusion section of the paper.

Round 3
Reviewer 1 Report
Dear authors,
the paper should be revised in order to publish it in the future
two reviews were given and according to them the paper can be sent to another journal with lower publishing criteria
Author Response
Thank you for your work.
In this modification, we have added 5.5 case extension analysis.
